# Polymerization Kinetics of Acrylic Photopolymer Loaded with Graphene-Based Nanomaterials for Additive Manufacturing

**DOI:** 10.3390/nano12244498

**Published:** 2022-12-19

**Authors:** Sara Lopez de Armentia, Juana Abenojar, Yolanda Ballesteros, Juan Carlos del Real, Nicholas Dunne, Eva Paz

**Affiliations:** 1Institute for Research in Technology, Mechanical Engineering Department, Universidad Pontificia Comillas, Alberto Aguilera 25, 28015 Madrid, Spain; 2Materials Science and Engineering and Chemical Engineering Department, Álvaro Alonso Barba Institute, Universidad Carlos III de Madrid, Av. Universidad 30, 28911 Leganés, Spain; 3Mechanical Engineering Department, Universidad Pontificia Comillas, Alberto Aguilera 25, 28015 Madrid, Spain; 4Centre for Medical Engineering Research, School of Mechanical and Manufacturing Engineering, Dublin City University, Stokes Building, Collins Avenue, Dublin 9, Ireland; 5School of Pharmacy, Queen’s University of Belfast, 97 Lisburn Road, Belfast BT9 7BL, UK; 6Department of Mechanical and Manufacturing Engineering, School of Engineering, Trinity College Dublin, Dublin 2, Ireland; 7Advanced Manufacturing Research Centre (I-Form), School of Mechanical and Manufacturing Engineering, Dublin City University, Stokes Building, Collins Avenue, Dublin 9, Ireland; 8Advanced Materials and Bioengineering Research Centre (AMBER), Trinity College Dublin, Dublin 2, Ireland; 9Advanced Processing Technology Research Centre, Dublin City University, Stokes Building, Collins Avenue, Dublin 9, Ireland; 10Trinity Centre for Biomedical Engineering, Trinity Biomedical Sciences Institute, Trinity College Dublin, Dublin 2, Ireland; 11Biodesign Europe, Dublin City University, Stokes Building, Collins Avenue, Dublin 9, Ireland

**Keywords:** Vat Photopolymerization, polymerization kinetics, graphene-based nanomaterials, acrylic-based resin

## Abstract

Graphene-based nanomaterials (GBN) can provide attractive properties to photocurable resins used in 3D printing technologies such as improved mechanical properties, electrical and thermal conductivity, and biological capabilities. However, the presence of GBN can affect the printing process (e.g., polymerization, dimensional stability, or accuracy), as well as compromising the quality of structures. In this study an acrylic photocurable resin was reinforced with GBN, using methyl methacrylate (MMA) to favor homogenous dispersion of the nanomaterials. The objective was to investigate the influence that the incorporation of GBN and MMA has on polymerization kinetics by Differential Scanning Calorimetry using Model Free Kinetics, ultra-violet (UV) and thermal triggered polymerization. It was found that MMA catalyzed polymerization reaction by increasing the chain’s mobility. In the case of GBNs, graphene demonstrated to inhibit both, thermally and UV triggered polymerization, whilst graphene oxide showed a double effect: it chemically inhibited the polymerization reaction during the initialization stage, but during the propagation stage it promoted the reaction. This study demonstrated that MMA can be used to achieve photocurable nanocomposites with homogenously dispersed GBN, and that the presence of GBN significantly modified the polymerization mechanism while an adaptation of the printing parameters is necessary in order to allow the printability of these nanocomposites.

## 1. Introduction

The addition of Graphene-Based Nanomaterials (GBN) to photocurable resins has been shown to produce improvements in different properties of the materials, e.g., mechanical [1,2,3], electrical [4,5,6], thermal [2,3,6] or biological [7] performance. One of the main challenges experienced during the preparation of these composites is that in order to exploit fully the potential of GBN it is necessary to ensure that they are homogenously dispersed within the resin. A common method used to achieve improved GBN dispersion is to add a low-viscosity solvent like water [8,9], toluene [6], acetone [10], isopropyl alcohol [11], or a mixture of isopropyl alcohol and butyl acetate [12,13].

During Vat Photopolymerization technologies (e.g., stereolithography [SLA], Digital Light Processing [DLP] and Liquid Crystal Display [LCD]) the polymerization of the resin occurs directly during the printing process. Therefore, when a solvent is added to improve dispersion of GBN it has to be properly removed. If there are traces of solvent in the mixture it can lead to a reduction in mechanical properties since it can affect the polymerization process impeding the crosslinking [14,15]. An alternative for the use of solvents is the dispersion of GBN into the liquid, low-viscosity monomers before curing [16,17]. However, the addition of monomers could affect polymerization kinetics of the resin for variety reasons, such as: (i) decrease in viscosity; (ii) change in components proportion; (iii) change in polymerization rate or mechanism. Therefore, it is fundamental to understand the effect that the auxiliary reactant used to improve GBN dispersion has over the fundamental resin properties.

Conversely, some studies have reported a reduction in mechanical properties of 3D printed components fabricated from a GBN-based composite material [10,18]. Manapat et al. [10] proposed four hypotheses: (i) the increasing concentration of GBN could produce excessive inter-platelet interaction instead of GBN–resin hydrogen bonding; (ii) GBN could act as barrier/hinderance to incoming laser light, reducing the efficiency of the photopolymerization process; (iii) GBN could inhibit polymerization due to it being a chain transfer agent; and (iv) the presence of wrinkles in the graphene sheets could affect the stress distribution, thereby hindering good adhesion between the graphene oxide and the resin. The effect of GBN on polymerization could be better understood by studying the polymerization kinetics of the resin in the presence of GBN.

The effect of different GBN on the thermal curing of different polymeric resins has been widely studied, and two different tendencies were found: (i) the presence of nanofillers accelerated the polymerization process by increasing thermal conductivity [19] or catalyzing the reactions due to the presence of oxygenated groups [20,21]; (ii) nanofillers reduced the polymerization reaction rate due to the steric hindrance that impeded the mobility of the reactants [19], or the increase in viscosity, which hindered the mobility of the reactive species [22]. To the best of our knowledge, few studies have been reported relating to acrylic-based photocurable resins. However, in the case of epoxy-based photocurable resins, nanofillers may reduce the polymerization reaction rate acting as radical scavengers, i.e., deactivating the photoinitiator, or due to its opaqueness against UV wavelength, which led to a reduction in the quantum efficiency. Additionally, the surface functional groups may also deteriorate the photo-generated species [4].

The photopolymerization of acrylic-based resins present three different stages [23]:

(i) Initialization: when the photoinitiator, in this case TPO (diphenyl(2,4,6-trimethylbenzoyl)phosphine oxide), absorbs UV light, free radicals are produced. These free radicals trigger the polymerization by promoting the homolytic cleavage of carbon bonds.

(ii) Propagation: the radical monomer reacts with more acrylate or methacrylate monomers and forms a radical oligomer. In this study there are urethane dimethacrylate (UDMA) and methacrylate monomers.

In general, the photopolymerization of acrylates leads to high crosslink densities because the process occurs in three dimensions, resulting in highly reticulated polymers. Their physical properties depend on the length and chemical structure of the crosslink segments [24].

(iii) Termination: it can occur via combination or disproportion. In the case of methacrylate monomers, the most probable pathway is disproportion [25].

The photopolymerization process is mainly governed by two parameters: the penetration depth of the curing light and the energy needed to polymerize [26]. It is important to notice that the power of light decreases when depth increases, following a Beer–Lambert relationship (Equation (1)):(1)$$P_{z}=P_{0}\cdot e^{-z/D_{p}}$$
where *P*_0_ and *P_z_* are the power of light (mW/cm^2^) at the surface and at some depth *z*, respectively, and *D_p_* is the depth where the penetrating light intensity falls 1/*e* of *P*_0_. Writing Equation (1) in terms of energy instead of power, Jacobs’ working curve equation (Equation (2)) is obtained [27].
(2)$$C_{d}=D_{p}\cdot ln\left[\frac{E_{0}}{E_{c}}\right]$$

*C_d_* being the cured depth, *E*_0_ the energy (mJ/cm^2^) of the light at the surface, *E_c_* the critical energy needed to start the polymerization and as it was previously defined, *D_p_* is the depth of penetration of the laser beam. This working curve is widely used in literature as a basic procedure for testing and characterizing photocurable resins [28,29,30,31].

To ensure sufficient interlaminar bonding, the actual curing depth must be larger than the layer thickness. If curing depth is not large enough delamination may occur and, therefore, printed structures present inferior properties [29]. To avoid delamination, exposure time could be increased to improve the curing depth [28]. However, over-curing could occur, leading to a detriment in printing accuracy.

In this study, acetone and methyl methacrylate (MMA) were proposed to be added to reduce the viscosity of the acrylic-based resin and improve the homogenous dispersion of the GBN. Firstly, the effect of the solvent on mechanical properties of the resin was evaluated to decide the solvent that presented the least negative effect.

The solvent that less affected mechanical properties of the acrylic-based resin was chosen and the effect of both, solvent and GBN, in thermally triggered and UV-triggered polymerization of an acrylic-based photopolymer was studied by means of non-isothermal differential scanning calorimetry (DSC). In the case of acrylic-based photopolymers, even without a thermal initiator, acrylate and methacrylate monomers have the ability to initiate polymerization when heat is applied [32]. The acrylic-based resin used in this study is made of UDMA and methacrylate monomers; therefore, in the case of thermally triggered tests the focus is on methacrylate monomers spontaneous self-initiated polymerization.

To the best of the authors’ knowledge, in the literature only hypotheses have been made regarding the effect of nanofillers on the properties and the polymerization of photocurable resins.

Therefore, the purpose of this research is to establish a foundation to understand the effect of GBN on the polymerization of photocurable resins. In addition, a solution to obtain homogeneous dispersion of GBN that does not affect negatively the polymerization process is proposed.

## 2. Materials and Methods

### 2.1. Materials

The resin used in this study was a photocurable acrylic-based resin (R), i.e., Formlabs Clear FLGPCL4 (Formlabs, Somerville, MA, USA) (Table 1).

FTIR spectrum of R can be seen in Figure 1. Bonds associated with each peak are shown in the image and correspond to the chemical composition of UDMA.

Two different GBN, from Avanzare Nanotechnology (La Rioja, Spain) and NanoInnova Technologies (Toledo, Spain) were used (Table 2).

### 2.2. Nanocomposites Preparation

The composition of the different GBN-based composites was prepared as shown in Table 3. Two solvents were explored: acetone and methyl methacrylate (MMA).

Samples with GBN were prepared as follows. R+MMA/Acetone samples were prepared following the same procedure, except the first step:

(i) GBN were dispersed in the solvent via probe ultrasonication using a Branson 450 ultrasonicator (Branson Ultrasonics Corp., CT., USA). The frequency range applied was 1985–2050 kHz at a 50% amplitude for 10 ± 0.5 min—pulses of 10 ± 0.5 s ON and 20 ± 0.5 s OFF and the solution was placed in an ice bath (6 ± 2 °C) to avoid the overheating;

(ii) Acrylic-based resin was added gradually to the previous solvent + GBN dispersion. The volume was doubled in each stage until the whole volume of resin (300 mL) was added. After each addition of resin, the same sonication cycle previously described (i) was applied;

(iii) Finally, degasification was undertaken in an ultrasonic bath (Elmasonic p60h, Elma Schmidbauer GmbH, Germany) for 15 ± 0.5 min, followed by a treatment in a vacuum drying oven (Vaciotem-T, Selecta, Spain) at room temperature (22 ± 0.5 °C) under vacuum (0.1 mbar) for 15 ± 0.5 min.

### 2.3. Printing of Tensile Samples

To choose the most adequate solvent, dog bone samples were fabricated using the mixtures R, R+MMA and R+Acetone, following standard ISO 527-2:2012, with SLA printer Form 2 (Formlabs, Somerville, MA, USA). Printing process was followed by cleaning in isopropyl alcohol for 3 min and postcuring was completed using the FormCure chamber (Formlabs, Somerville, MA, USA) at 80 °C for 90 min.

### 2.4. Absorbance

Absorbance of the liquid samples was measured at 405 nm with a UV-Vis Cary 4000 spectrophotometer (Agilent Technologies, Santa Clara, CA, USA). The wavelength was selected as a function of the wavelength of the printer’s laser. As reference, R sample was used. This parameter allowed the determination of the effect of GBN on light absorbance, which could influence the UV polymerization process.

### 2.5. Jacobs’ Working Curve

The power (*P*_0_) of the UV lamp used was 358 mW/cm^2^. Using this information it is possible to calculate the energy for different exposure times (Equation (3)).
(3)$$E_{0}=P_{0}\cdot t$$

From Jacobs’ working curve equation (Equation (2)), it is possible to see that if *C_d_* vs. *ln*(*E*_0_) is plotted, *D_p_* and *E_c_* can be determined. *D_p_* corresponded with the slope of the line, whilst *E_c_* was calculated from the intersection with x-axis, knowing that it is plot as the natural logarithm.

The liquid samples were exposed to UV light at different exposure times: 3, 5, 10 and 15 s and then the solid film formed on the surface was cleaned using the FormWash (Formlabs, Somerville, MA, USA) for 3 min to remove uncured resin. Once cleaned, the thickness was measured with a Vernier caliper to an accuracy of 0.01 mm. 

At least six measurements were completed for each sample and the average and standard deviation was calculated. In all cases, the coefficient of determination (R^2^) of the regression line was higher than 0.99.

Volume efficiency, *γ_V_*, was used to evaluate the effectiveness of the polymerization, and was calculated using Equation (4) [33]:(4)$$\gamma_{V}=\frac{D_{p}}{E_{c}}$$

### 2.6. Polymerization Degree

Samples subjected to 2 s of UV were analyzed with a Tensor27 FTIR spectrometer (Bruker Optik GmbH, Madrid, Spain) with DuraSample Diamond accessory formed by a 0.5 mm diameter diamond embedded in a ZnSe crystal; this was attached to the spectrophotometer and attenuated total reflectance (ATR) technique was used. The ratio signal-to-noise is better than 8000:1 (5.4 × 10^−5^ noise absorbance). Spectra were recorded with a resolution of 4 cm^−1^ from 4000 to 400 cm ^−1^ by taking 32 scans. FTIR spectra were recorded and analyzed with OPUS software (Bruker Optik GmbH, Madrid, Spain). The objective of this test is to analyze polymerization degree of UV-triggered polymerized samples before carrying out DSC scans.

Polymerization degree was calculated from the peaks corresponding to C=O and C=C bonds, located at 1725 and 810 cm^−1^ [34] (Equation (5)). As reference, liquid resin with and without GBN was used.
(5)$$X=\left[1-\frac{I_{\left(C=C\right)}}{I_{liquid\left(C=C\right)}}\cdot \frac{I_{liquid\left(C=O\right)}}{I_{\left(C=O\right)}}\right]\cdot 100$$

### 2.7. Thermally Triggered Polymerization

Once the sample was prepared, approx. 25–30 mg of the sample was placed in an aluminum crucible with a capacity of 40 µL and a 50 µm hole in the lid. DSC 882e Mettler Toledo (Greifensee, Switzerland) was used to complete scans from 20 to 250 °C at three different rates (i.e., 5, 10 and 20 °C/min). Nitrogen was used as the purge gas and was delivered at a rate of 80 mL/min. Each scan was carried out at least three times.

Activation energy (*E_a_*) of the polymerization process was evaluated by model free kinetic (MFK) method, applied with STARe Software (Mettler Toledo, Greifensee, Switzerland). Firstly, conversion degree (α) was calculated from each of the obtained curves at the three different rates. From these values, *E_a_* was calculated as a function of α. It is important to highlight that activation energy calculated by MFK method changes with the extent of the polymerization. Therefore, it is possible to distinguish between the different stages of the polymerization reaction, as previously found by Paz et al. [21]. The model used by STARe software is based on the work of Vyazovkin et al. [35].

### 2.8. UV-Triggered Polymerization

In this case, prior to DSC scan samples were subjected to 405 nm UV light for 2 s of exposure time. It was done with a UV LED lamp of 358 mW/cm^2^ of light power supplied by Sovol (Shenzhen, Guangdong, China). After this exposition that triggered polymerization reaction, the same procedure explained above was followed to determine kinetic parameters.

### 2.9. Glass Transition Temperature

The glass transition temperature (T_g_) was obtained from each of the DSC spectra. T_g_ of thermally triggered and UV-triggered samples was calculated. In the first case, liquid resin was placed in the crucible. In the case of the UV-triggered sample, a drop of resin was placed in the crucible, and it was subjected to 2 s of UV light, as previously explained.

Two scans from 20 to 250 °C were carried out at 20 °C/min. In the first scan, the sample was completely cured and T_g_ was read in the second scan. It was calculated as the midpoint of the step of the base line through endothermal direction.

### 2.10. Statistical Analysis

One-way analysis of variance (ANOVA) test with a post hoc Scheffe’s test was used to evaluate the results for statistical significance with the software SPSS 20.0 for Windows (IBM SPSS, Chicago, IL, USA). A *p*-value lower than 0.05 was indicative of statistical significance.

The scheme of the experimental process can be seen in Figure 2.

## 3. Results

### 3.1. Effect of Solvents on the Mechanical Properties

Firstly, the effects of the solvents on the mechanical properties of the acrylic-based resin were studied. In Figure 3, the normalized values for the tensile strength and Young’s moduli values are shown. The values of R were taken as a reference: 34.5 MPa and 1.3 GPa, respectively.

The addition of acetone and MMA had no significant effect on Young’s modulus of the acrylic-based resin, with *p*-values of 0.910 and 0.726, respectively. However, it was observed that the addition of MMA produced a non-significant (*p*-value = 0.131) decrease in tensile strength of 12%. Meanwhile, the tensile strength of the R+Acetone was approx. 70% of the tensile strength of the resin (*p*-value = 0.001). This could be due to the presence of small traces of acetone due to the evaporation process being incomplete or because the acetone degraded or changed the reactant structure responsible for the polymerization. Considering these results, acetone was discarded as a solvent to reduce the resin viscosity and only MMA was used to add the GBN. From now on, R+MMA will be referred to as R’.

### 3.2. Absorbance

Table 4 shows the absorbance of the different samples measured using R as a reference.

The addition of MMA to the acrylic-based resin did not produce a variation in the absorbance of the sample. However, the addition of GBN produced an important change in this parameter. The addition of graphene showed a more notable increase in the absorbance when compared to graphene oxide. This increase could affect the penetration depth of the light and, therefore, the polymerization of the resin by UV.

### 3.3. Jacobs’ Working Curve

To study the effect of GBN on the penetration of the light, Jacobs’ working curve of the different nanocomposites was obtained (Figure 4).

R’ showed the highest values of curing depth, whilst R’+G showed the lowest. In the case of R’+GO, the curing depth was similar to R. From Jacobs’ working curves, the critical energy (*E_c_*), depth of penetration (*D_p_*) and volume efficiency (*γ_V_*) were obtained applying Equations (2) and (5) (Table 5). These two parameters directly affected the *C_d_*. The higher the *E_c_* and/or the lower the *D_p_*, the lower the *C_d_*.

The addition of MMA did not affect depth of penetration for the acrylic-based resin. However, critical energy to trigger the polymerization was reduced.

When GBN were added, both parameters changed. Comparing to R, penetration depth was reduced, as well as critical energy. This effect could be due to the high absorbance of R’+GBN because of the change in the color of the resin. However, when R’ was taken as reference, the effect of GBN was different—graphene produced an increase in *E_c_* and a decrease in *D_p_*, whilst graphene oxide reduced both parameters.

The reduction in *E_c_* observed when graphene oxide was added could suggest a catalytic effect of this GBN on the polymerization reaction. The decrease in *D_p_* observed for both GBN could be due to the absorbance of these GBN that led to a reduction in the efficiency of light exposure in terms of free radical generation. This decrease was more pronounced in the case of the graphene, being consistent with the results observed in the absorbance measurements.

### 3.4. Thermally Triggered Polymerization

All the thermograms obtained showed one exothermic peak, which corresponds to the radical polymerization of the resin. The asymmetry of the polymerization peaks and the possible appearance of two different peaks in some nanocomposites and rates could be due to the different rates at the different stages due to the auto-acceleration in the propagation stage [36]. They can be seen in Appendix A.

In this part of the study, TPO (the photoinitiator) did not produce free radicals because the UV did not trigger its decomposition. Therefore, R polymerization occurred via spontaneous self-initiated thermal polymerization of methacrylate monomers, which was triggered at high temperatures (100–130 °C). In general, it was found that this reaction resulted in a low conversion of monomers to polymers and high conversion to oligomers (dimers or trimers) [32].

#### 3.4.1. Effect of MMA Addition

The addition of MMA to the resin produced a reduction of 54% in the energy released during polymerization (Table 6). A ΔH decrease means that MMA promoted and facilitated the polymerization reaction [37]. In terms of peak temperature, it showed similar values for both samples at every rate.

The polymerization reaction of R’ started at a lower temperature than R (Figure 5). However, once 50% of conversion was reached R’ required higher temperatures to continue its polymerization. Additionally, the slope of the conversion curve changed slightly on MMA addition, which could be attributed to a change in polymerization rate. The addition of monomer led to a formation of more radicals, which resulted in a decrease in the initial polymerization temperature. Each curve was obtained three times and the error was estimated to be lower than 1%.

On the contrary to R, analyzing the activation energy of R’ reaction (Figure 6) it was observed that during the initialization stage the values or *E_a_* are notably lower than propagation and termination stages, which means that MMA favored the initialization of the reaction. This was probably due to the decrease in viscosity, which resulted in a higher mobility of chains. Additionally, with the addition of MMA there was a higher probability of occurrence of spontaneous self-initiated thermal polymerization of MMA compared to larger acrylate monomers (e.g., UDMA).

Once 5–10% of conversion was achieved, the *E_a_* of R’ was higher because of the increase in viscosity due to the formation of a three-dimensional crosslinked network. Finally, in the termination stage, from around 85% of conversion *E_a_* in both cases increased because this stage of the reaction was diffusion-controlled due to the high polymerization degree and the restriction of the chains’ mobility [38].

#### 3.4.2. Effect of GBN Addition

Thermograms can be seen in Appendix A. Enthalpy, conversion curves and activation energy were calculated from them.

GBN showed a decrease in the polymerization enthalpy (Table 7). This effect was more pronounced for graphene. In all cases, peak temperature was higher than control sample (R’). This increase in the peak temperature showed a retardation effect of GBN in the polymerization reaction [39]. R’+G showed the most pronounced increase in temperature and the lower ΔH; therefore, it is possible that R’+G reached a lower crosslinking or polymerization degree.

In the conversion curves shown in Figure 7, a shift to higher temperatures of the curve produced by GBN was clearly shown. In addition, a change in the slope of the R’+GO curve could indicate a change in polymerization rate towards a faster mechanism, where graphene oxide acts as reaction catalyzer.

It can be observed that graphene and graphene oxide had a different effect on *E_a_* during the polymerization (Figure 8). The addition of graphene resulted in higher activation energy in all the conversion range, and it could be attributed to the increase in viscosity and the hindering of the chains’ mobility produced by the presence of graphene [39]. However, although the addition of graphene oxide also produced this increase in viscosity, in propagation and termination stages the activation energy needed to complete the reaction was lower than R’. This could be explained by a catalytic effect of the graphene oxide. Conversely, despite this catalytic effect during the last stages, it is observed that at initialization stage, graphene oxide inhibited the self-initiated polymerization. This effect will be discussed later and is attributed to the oxygenated presents on the graphene oxide surface.

### 3.5. UV-Triggered Polymerization

This study was focused on how the reaction proceeded when the resin was triggered by UV; however, that initial UV polymerization was not monitored by DSC. Therefore, polymerization degree of UV-exposed samples was calculated by FTIR analysis before carrying out DSC scans.

Results obtained from FTIR spectra applying Equation (5) are shown in Table 8.

It can be observed that the addition of MMA greatly increased polymerization degree whilst when GBN were added to R’ the polymerization degree decreased, especially in the case of G. Therefore, the initial polymerization degree of R’ was higher than the other materials.

#### 3.5.1. Effect of MMA Addition

Table 9 shows the analysis of the polymerization peak obtained for the thermal polymerization of samples previously exposed to UV for 2 s. Thermograms can be seen in Appendix A.

In comparison with thermal polymerization, when polymerization was triggered by UV the effect of the addition of MMA drastically changed. In this case, the area under the polymerization peak suffered an increase of 74% and the temperature decreased. This reduction in temperature suggests that this composite with MMA polymerized by heat more readily following UV when compared to R, and the increase in enthalpy suggests that the crosslinking degree is higher than R. From the DSC spectra, conversion curves of these samples were obtained (Figure 9).

The addition of MMA produced a shift in the curve towards a lower temperature. However, the slope of the curve was the same, meaning that the polymerization rate did not change. In this case, the photoinitiator was the source of free radicals in both cases, and therefore the presence of MMA did not change the polymerization rate or mechanism. From conversion curves, activation energy was calculated for each residual polymerization degree. The resultant curves are shown in Figure 10.

At the beginning, both curves showed low activation energy because the reaction was initiated by UV, and therefore, there were free radicals available to polymerize. However, it could be seen how the activation energy of R was higher than R’ for every conversion degree, which was consistent with the catalytic effect due to the decrease in viscosity previously commented. This could be explained by the higher viscosity of R compared to R’ and the subsequent restriction in the chains’ mobility. Additionally, R’ had a higher concentration of monomers available, which could result in a lower *E_a_*.

#### 3.5.2. Effect of GBN Addition

Table 10 shows the residual enthalpy and the temperature of the polymerization peak of the nanocomposites after UV exposition. Thermograms can be seen in Appendix A.

As reported previously in thermally triggered polymerization, GBN reduced the energy and increased the temperature. The increase in temperature explains that GBN retarded polymerization, together with the decrease in enthalpy, suggests a decrease in crosslinking degree.

The representation of the residual conversion after the UV initialization is shown in Figure 11. The addition of graphene did not change the slope of the conversion curves. However, the addition of graphene oxide slightly increased the slope. This means that the addition of graphene oxide modified the polymerization rate, probably due to its catalytic effect which was also observed in Figure 7.

In relation to the activation energy, Figure 12 reported a similar trend to the thermally initiated polymerization. In the case of graphene oxide at the initial stage, it inhibited the polymerization. Once a certain conversion level was reached this effect changed, and graphene oxide catalyzed the reaction. In the case of R’+G mixture the exposure to UV produced fewer free radicals, as graphene had a very high absorbance and did not allow the whole light to reach the photoinitiator. For this reason, its polymerization enthalpy was lower, and its activation energy was higher.

The trend found for thermally and UV-triggered polymerization was different, and they are compared in the discussion section.

### 3.6. Glass Transition Temperature

Table 11 reports the glass transition temperatures of the different samples. The polymerization was triggered by heat and by UV. The thermograms used to measure glass transition temperature are shown in Appendix A. They show the successful polymerization in all cases.

Differences found were within the error of DSC measurements. Therefore, the differences in polymerization did not result in changes in the T_g_.

## 4. Discussion

In this study, the kinetics of polymerization initiated by UV and heat were studied in detail. To understand and explain the results, some additional measurements were completed, i.e., absorbance, penetration depth and polymerization degree obtained by UV exposition.

Firstly, to improve the homogenous dispersion of the GBN two solvents were explored: acetone and MMA. Acetone showed a decrease in tensile properties due to the incomplete evaporation of the solvent or a degradation or change in the polymer structure caused by acetone [15]. Conversely, MMA did not need to be evaporated because it could polymerize with the acrylic-based resin without significantly affecting the mechanical properties.

The increase in absorbance also had an effect in the penetration depth of light. It was found that the presence of graphene and graphene oxide produced a reduction in *D_p_* (i.e., 29.6% and 22.8%). The planar shape of GBN may block light scattering throughout the resin [40], resulting in lower penetration depth. In the case of critical energy it was found that the addition of MMA produced a reduction of 27.6% for this parameter. Compared to R’, graphene oxide produced a reduction of 28.8% on *E_c_* and graphene did not change it.

The MMA effect (decrease in *E_c_* maintaining *D_p_*) has been previously reported by Hofstetter et al. [29] when increasing the photoinitiator content. Therefore, the effect of MMA could be similar to the effect of an increase in photoinitiator amount and promoted the UV polymerization, as can be seen in the increase in *γ_V_*. It could be due to the decrease in viscosity, which increased the chains’ mobility and facilitated the light to reach the photoinitiator, increasing the UV polymerization efficiency. In the case of GBN, results suggest that graphene only affected penetration depth, whilst graphene oxide catalyzed polymerization due to its oxygenated groups. It was previously found [21] that graphene and graphene oxide produced a catalytic effect on the polymerization of the acrylic-based resin, being the effect of graphene oxide more marked than that of graphene due to its higher levels of functionalization. In this case, graphene produced a reduction in *γ_V_*, whilst graphene oxide increased this parameter due to its catalytic effect.

Polymerization degree measured using FTIR spectroscopy reinforced the conclusions extracted from Jacobs’ working curves. The higher polymerization degree was found for R’, which showed the lower *E_c_* with the same *D_p_* as R. In the case of R’+G and R’+GO, their *E_c_* are also lower than R, but the decrease in *D_p_* resulted in lower polymerization degree because the light effect was hampered by the presence of GBN.

However, Jacobs’ working curves give little information about polymerization kinetics [29]. For this reason, it is necessary to broaden the study to completely characterize the polymerization process of nanocomposites.

To analyze polymerization kinetics, it is important to highlight some aspects:−It can be appreciated that all curves showed a sigmoidal form, which indicated that the polymerization reaction was autocatalytic, as corresponds to free-radical polymerization of acrylics [22];−In general, a change in the heat released during polymerization may be due to two causes [21]: (i) the degree of polymerization has changed; and (ii) the polymerization mechanism or rate has been modified. It is possible to know if the reaction mechanism and the polymerization parameters are the same by observing the conversion vs. temperature curve. Its shape and slope are related to the polymerization mechanism and rate; therefore, a change in its shape and/or its slope indicates a change in the mechanism and the parameters [41,42];−In the case of resin with GBN, the decrease in enthalpy shown could be due to two reasons [40]: (i) heat generated during the polymerization could be transferred to GBN, which would result in a decrease in enthalpy; (ii) the planar shape of GBN could block light scattering throughout the resin, impeding, totally or partially, the polymerization.

During thermally initiated polymerization, when MMA is added ΔH decreased and the reaction started at lower temperatures in the conversion vs. temperature curve. Therefore, the initialization of the polymerization reaction was catalyzed by the presence of MMA. It could be explained by three events: (i) the concentration of monomer was higher; (ii) the viscosity was lower; and (iii) MMA is the smallest methacrylate monomer with the higher mobility. The combination of these facts made the concentration and the mobility of reactants increase in the first stage of the polymerization [43].

To facilitate the comparison and the discussion of the polymerization kinetics, the activation energy required for each stage of the polymerization is summarized in Table 12.

The effect of MMA was also found in activation energy (Table 12), especially in the initialization stage. In addition, R’ presents a broader range of temperature to complete its polymerization (Figure 5). This could indicate that a less homogeneous network is formed during the reaction [44] because UDMA and MMA could generate different networks.

The addition of GBN also produced changes in polymerization kinetics, but its effect was different depending on the nature of the GBN. The addition of graphene did not change polymerization mechanism or rate. Therefore, the decrease in enthalpy observed in Table 8 could be due to a decrease in crosslinking or polymerization degree. In addition, there was an increase in temperature which showed a retardation in the polymerization [39]. The extent of the reaction was lower because graphene had high absorbance and the lowest value of *γ_V_*, which could cause steric hindrance and did not allow the polymerization to occur properly. This effect was also found in the increase in activation energy compared to R’ in all the stages (Figure 8).

In the case of graphene oxide its effect was different. In Figure 7, a modification on the polymerization rate can be seen. It could be due to the catalytic effect of the high concentration of oxygenated groups that graphene oxide presented, which was also found in the decrease in enthalpy [37]. However, this effect was not shown at the beginning of the reaction where graphene oxide inhibited polymerization acting as radical scavenger, probably due to its oxygenated groups [4]. These groups are electrophiles, and they can react with nucleophile carbons of methacrylate monomers, inhibiting this first stage of the reaction.

When polymerization was initiated by UV exposure, the reaction completion by heat was studied by DSC. It is important to highlight that in this case the initial conversion was not 0 since some polymerization occurred during the UV exposure. This previous polymerization was not monitored, but as explained above, polymerization degree was obtained by FTIR spectroscopy (Table 6). It can be seen that conversion vs. temperature curves of R and R’ presented a similar slope (Figure 9). Therefore, a change in polymerization rate did not occur. R’ demonstrated higher initial polymerization degree than R, and for this reason, the polymerization released more energy.

In the case of R’+G, it was previously found by absorbance measurements (Table 4) and Jacobs’ working curve (Figure 4) that light could not reach the photoinitiator because of the presence of G. For this reason, there were fewer free radicals present in the mixture and the polymerization progress was retarded. It was shown in a lower enthalpy, higher polymerization temperature and higher activation energy. When graphene oxide was added in the first stage it seemed that graphene oxide acted as radical scavenger due to the phenolic hydroxyl or carboxyl groups on its surface. These groups may react with the initiator primary radicals by hydrogen abstraction. This resulted in a reduction in the initiator efficiency and, therefore, a retardation of the initialization stage of the reaction and an increase in activation energy [45] (Table 12). When the reaction progressed, the functional groups on the surface of graphene oxide were not free and the scavenger effect disappeared.

If the released enthalpy of thermally triggered polymerization is compared to UV-triggered polymerization, different trends are found. At this point, it is important to highlight again that a UV-triggered initialization stage is not the real initialization because the formation of free radicals was done prior to these tests, and it was not monitored. In all cases (except R), it was found that the enthalpy for UV-triggered polymerization was higher than for the reaction triggered by heat. It could be explained by the initial polymerization degree that involved an increase in viscosity and, therefore, an increase in activation energy. This trend was not found for R since it did not have MMA in its composition and, therefore, the spontaneous self-initiated polymerization probably did not occur to the same extent.

Finally, no significant changes were observed in T_g_ values, which suggests that despite the changes produced by MMA and GBN on the polymerization process the resulting polymer was not affected in terms of glass transition.

## 5. Conclusions

Some research papers unexpectedly found that GBN produced a decrease in mechanical properties, which could be due to the presence of agglomerates or an effect of GBN on polymerization process. This study focused on the effect of MMA and GBN on the polymerization reaction of an acrylic-based photopolymer when this reaction has been triggered by heat or UV. The addition of MMA produced improvements in dispersibility by reducing resin viscosity without producing significant effects on mechanical properties. Its presence favors both thermal and UV polymerization. In the case of graphene, it inhibited thermal and UV polymerization, as was observed in the increase in activation energy and in Jacobs’ working curve. When it was added to the resin it hindered the penetration of UV light, resulting in lower volume efficiency. The graphene oxide showed a double effect. On one hand, it chemically inhibited the polymerization reaction during the initialization stage but, conversely, during the propagation stage the R’+GO showed a higher polymerization rate than R’. Therefore, it showed a catalytic effect during this stage. In the case of UV polymerization graphene oxide produced a decrease in *D_p_*, but it was counterbalanced by the catalytic effect resulting in higher volume efficiency. Therefore, this research study demonstrated that graphene negatively affected the polymerization process and printing parameters must be optimized, while graphene oxide favored the polymerization without presenting any negative effect. In addition, MMA can be used to get nanocomposites with homogenously dispersed nanofillers and acrylic-based resin as matrix, which will allow taking advantage of the improvements of nanofillers together with the good performance of Additive Manufacturing.

## Figures and Tables

**Figure 1 nanomaterials-12-04498-f001:**
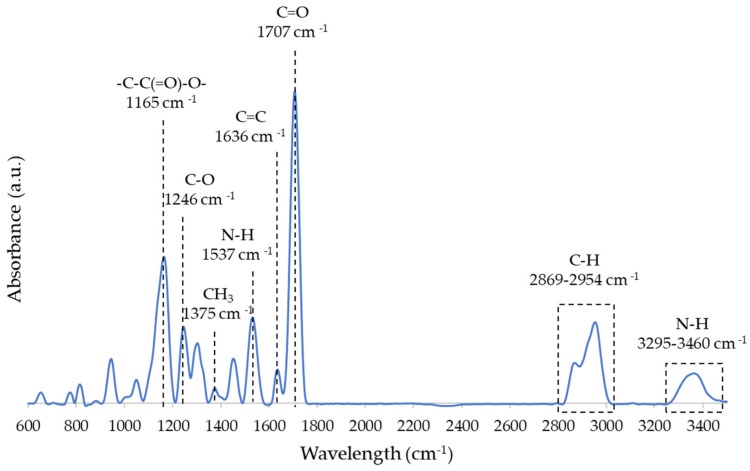
FTIR spectrum of acrylic resin.

**Figure 2 nanomaterials-12-04498-f002:**
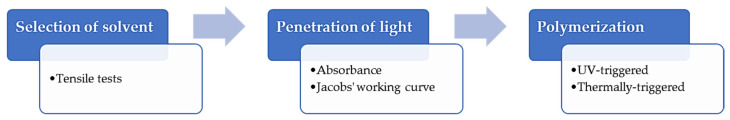
Experimental procedure.

**Figure 3 nanomaterials-12-04498-f003:**
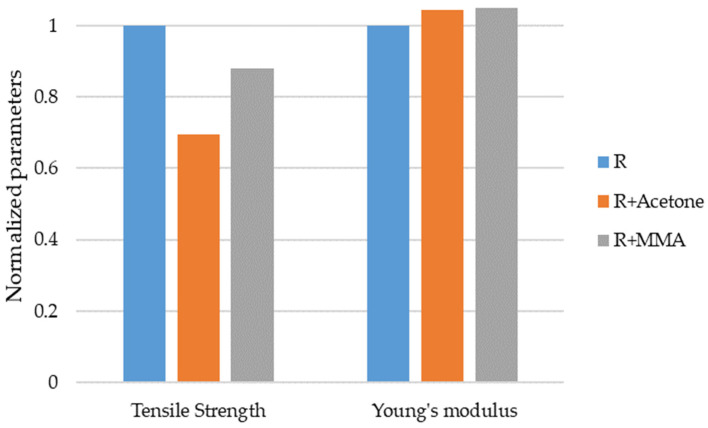
Effect of acetone and MMA on tensile properties of the resin.

**Figure 4 nanomaterials-12-04498-f004:**
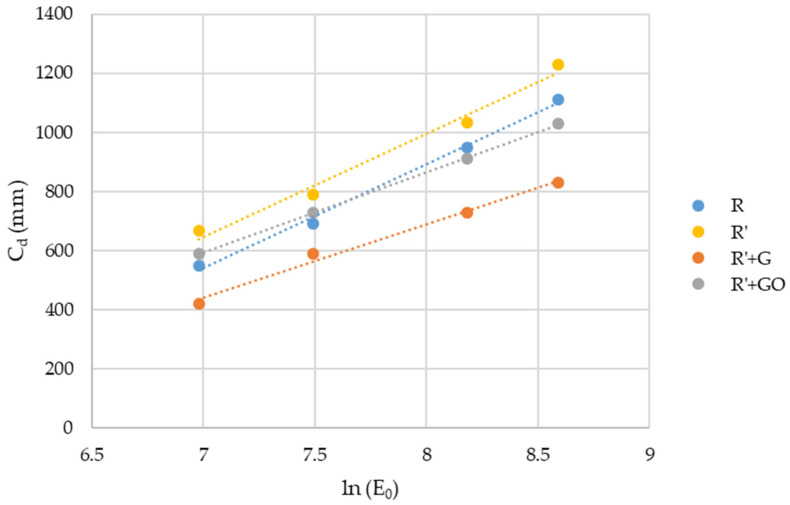
Jacobs’ working curve of R, R’ and R’+GBN.

**Figure 5 nanomaterials-12-04498-f005:**
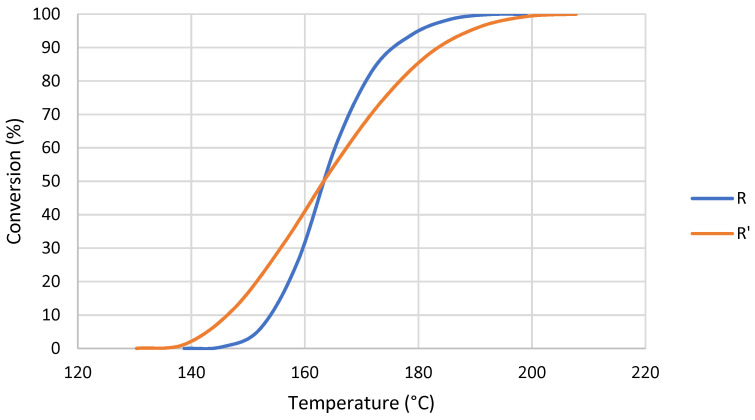
Conversion curves at 10 °C/min of R and R’.

**Figure 6 nanomaterials-12-04498-f006:**
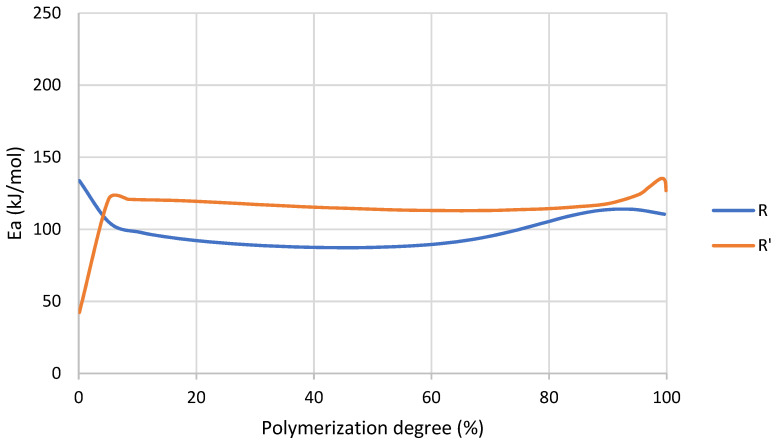
Activation energy as a function of conversion degree of R and R’ obtained by MFK model.

**Figure 7 nanomaterials-12-04498-f007:**
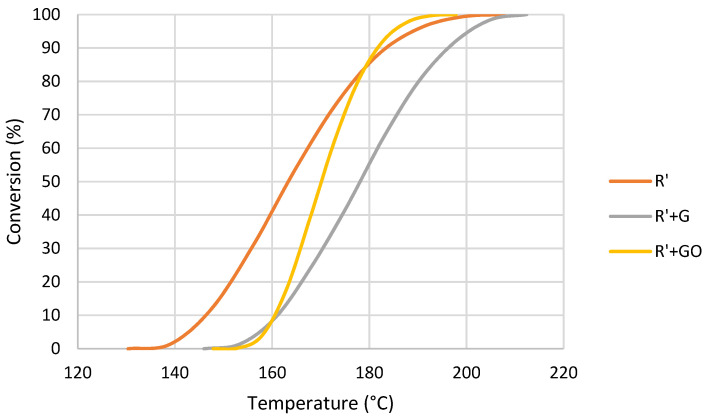
Conversion curves at 10 °C/min of R’ and R’+GBN.

**Figure 8 nanomaterials-12-04498-f008:**
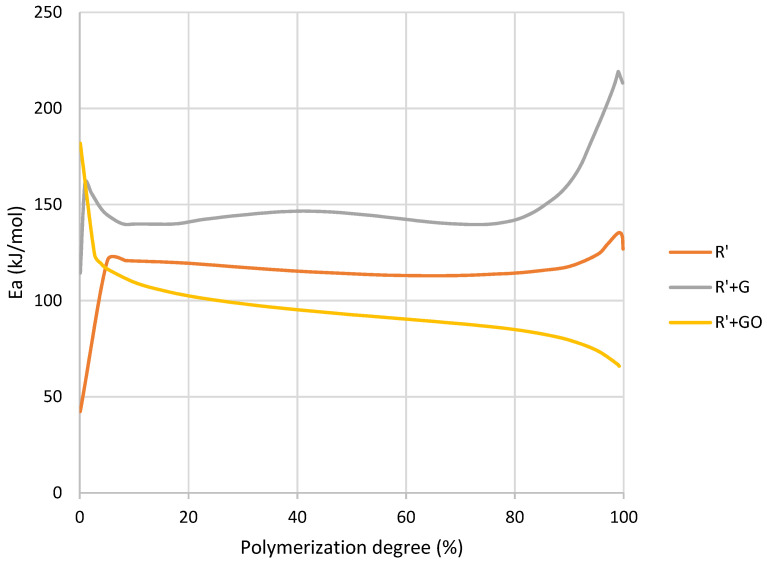
Activation energy as a function of conversion degree of R’ and R’+GBN obtained by MFK model.

**Figure 9 nanomaterials-12-04498-f009:**
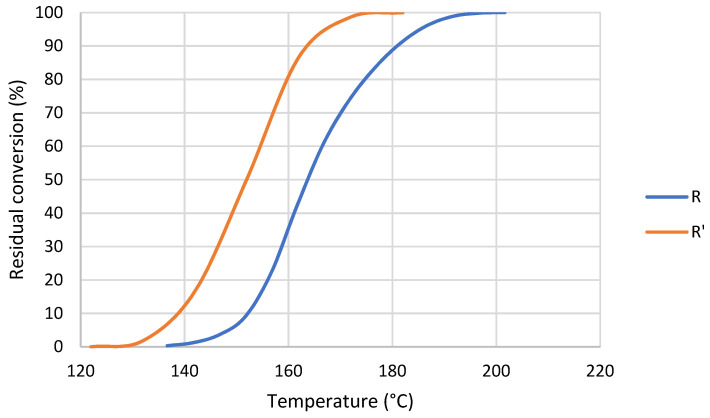
Conversion curves at 10 °C/min of R and R’ after UV exposure.

**Figure 10 nanomaterials-12-04498-f010:**
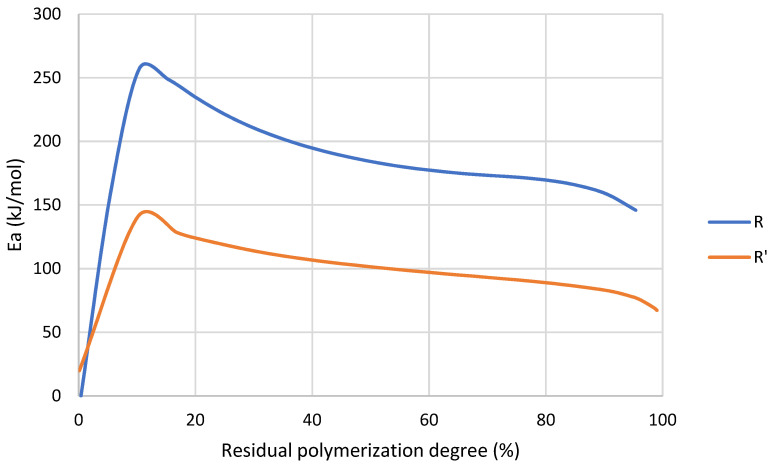
Activation energy as a function of residual conversion degree of R and R’ after UV exposure obtained by MFK model.

**Figure 11 nanomaterials-12-04498-f011:**
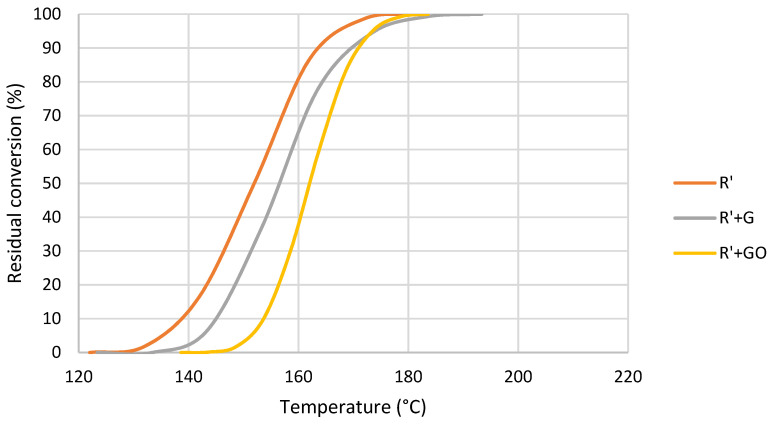
Conversion curves at 10 °C/min of R’ and R’+GBN after UV exposure.

**Figure 12 nanomaterials-12-04498-f012:**
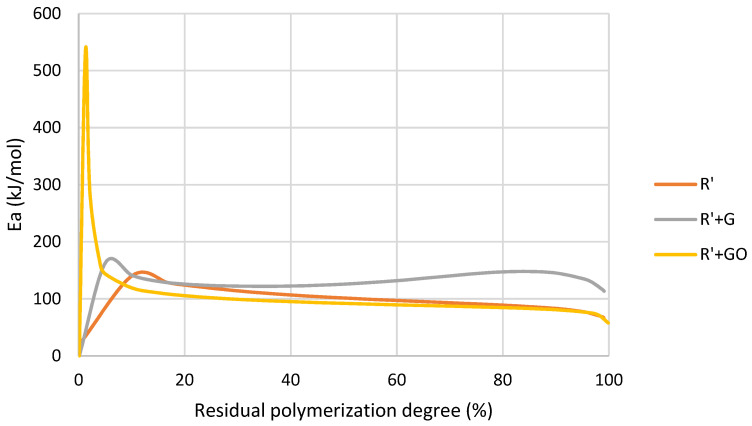
Activation energy as a function of residual conversion degree of R’ and R’+GBN after UV exposure obtained by MFK model.

**Table 1 nanomaterials-12-04498-t001:** Formlabs Clear V4 resin composition according to supplier data sheet.

*Component*	*Weight %*
Urethane dimethacrylate (UDMA)	55–75
Methacrylate Monomer(s)	15–25
Diphenyl(2,4,6-trimethylbenzoyl)phosphine oxide (TPO)	<0.9

**Table 2 nanomaterials-12-04498-t002:** GBN characteristics according to supplier data sheet.

*GBN*	*Average Lateral Size (µm)*	*Thickness*	*Supplier*
Graphene (G)	2–4	1–2 sheets	Avanzare Nanotechnology (La Rioja, Spain)
Graphene Oxide (GO)	4–8	0.7–1.2 nm	NanoInnova Technologies (Toledo, Spain)

**Table 3 nanomaterials-12-04498-t003:** Composition of GBN-based composite samples.

*Sample*	*R:Solvent Ratio*	*GBN Content*
R	-	-
R+MMA/Acetone (R’)	300:10 *v/v*	-
R’+G/GO	300:10 *v/v*	0.05 wt.%

**Table 4 nanomaterials-12-04498-t004:** Absorbance measured at 405 nm.

	*Absorbance (a.u.)*
**R’**	0.02
**R’+G**	1.62
**R’+GO**	1.00

**Table 5 nanomaterials-12-04498-t005:** *E_c_* and *D_p_* obtained from Jacobs’ working curves of acrylic-based resin and its nanocomposites.

	*E_c_* (mJ/cm^2^)	*D_p_* (µm)	*γ_V_* (mm^3^/J)
**R**	235	351	149
**R’**	170	347	204
**R’+G**	183	247	135
**R’+GO**	121	271	223

**Table 6 nanomaterials-12-04498-t006:** Polymerization enthalpy and curing peak temperature of R and R’.

	*Δ**H_polymerization_* (J/g)	*T_p_* (°C) *at Different Rates* (°C/min)		
		*5*	*10*	*20*
* **R** *	*122 ± 3*	*151*	*162*	*176*
* **R’** *	*56 ± 2*	*152*	*160*	*174*

**Table 7 nanomaterials-12-04498-t007:** Polymerization enthalpy and curing peak of R’ and R’+GBN.

	*Δ**H_polymerization_* (J/g)	*T_p_* (°C) *at Different Rates* (°C/min)		
		*5*	*10*	*20*
* **R’** *	*56 ± 2*	*152*	*160*	*174*
* **R’+G** *	*29 ± 2*	*169*	*178*	*181*
* **R’+GO** *	*52 ± 1*	*159*	*168*	*181*

**Table 8 nanomaterials-12-04498-t008:** Degree of polymerization of samples subjected to UV for 2 s.

	*X* (%)
**R**	8.42
**R’**	41.03
**R’+G**	2.87
**R’+GO**	6.96

**Table 9 nanomaterials-12-04498-t009:** Residual enthalpy and curing peak of R and R’ after UV exposure.

	*Δ**H_residual_* (J/g)	*T_p_* (°C) *at Different Rates* (°C/min)		
		*5*	*10*	*20*
** *R* **	*65 ± 3*	*157*	*160*	*169*
** *R’* **	*117 ± 1*	*140*	*153*	*160*

**Table 10 nanomaterials-12-04498-t010:** Residual enthalpy and curing peak of R’ and R’+GBN after UV exposure.

	*Δ**H_residual_* (J/g)	*T_p_* (°C) *at Different Rates* (°C/min)		
		*5*	*10*	*20*
** *R’* **	*117 ± 1*	*140*	*153*	*160*
** *R’+G* **	*56 ± 2*	*155*	*164*	*176*
** *R’+GO* **	*81 ± 2*	*154*	*161*	*176*

**Table 11 nanomaterials-12-04498-t011:** Glass transition temperature of samples with thermally and UV-triggered polymerization.

		*T_g_* (°C)
R	Thermal	209 ± 2
	UV	208 ± 1
R’	Thermal	206 ± 3
	UV	202 ± 3
R’+G	Thermal	208 ± 2
	UV	210 ± 3
R’+GO	Thermal	207 ± 2
	UV	210 ± 4

**Table 12 nanomaterials-12-04498-t012:** Activation energy (kJ/mol) of thermally and UV-triggered resins.

		*Initialization (α = 2%)*	*Propagation (α = 50%)*	*Termination (α = 90%)*
R	Thermal	100	88	113
	UV	79	184	159
R’	Thermal	82	114	118
	UV	92	101	83
R’+G	Thermal	156	145	162
	UV	78	126	145
R’+GO	Thermal	114	93	79
	UV	535	92	81

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
