# Peer review of "Polymerization Kinetics of Acrylic Photopolymer Loaded with Graphene-Based Nanomaterials for Additive Manufacturing"

_nanomaterials, 2022, doi:10.3390/nano12244498_

Round 1
Author Response
Dear Editor and Reviewer:
We are sending you the revised version of the manuscript entitled:
"Polymerization Kinetics of Acrylic Photopolymer loaded with Graphene-Based Nanomaterials for Additive Manufacturing” (Manuscript ID: nanomaterials-2034991), prepared by S. Lopez de Armentia, J. Abenojar, Y. Ballesteros, J.C. del Real, N. Dunne, E. Paz.
We thank the reviewers for their careful reading of the manuscript and their valuable comments that have helped us to improve it.
We hope to have answered all the queries raised by the reviewer and the editor and that the manuscript is now acceptable for publication in Nanomaterials. The reviewer’s comments followed by the corresponding answers are shown as follows:
The current work entitled Polymerization Kinetics of Acrylic Photopolymer loaded with Graphene Based Nanomaterials for Additive Manufacturing, is a research with considerable experimental work associated to the polymerization kinetics of an acrylic photocurable resin, before and after reinforcement with graphene based nanomaterials (GBN) and under the effect of MMA addition. The study could be published after considering the following comments:
- Table1: When 55% of UDMA is used with e.g. 25% of Methacrylate Monomers the total does not result in 100% for the system. What do the authors imply by these mixture quantities?
The quantities of the different components of the resin were given by the resin manufacturer. Due to secret trade, they did not give us the exact percentages and it is only an approximate composition. We agree with you, but it is the information given by the manufacturer.
- DSC curves (heat flow vs temperature) should be included for the tested samples (the first and second run) under all estimated different heating rates (5,10 and 20oC/min).
DSC curves have been added as supplementary files.
- Why did the authors select crucibles with a hole and not hermetically sealed? Does any substance evaporate from the mixture? It would be helpful if a mass loss curve vs temperature could be depicted through Thermogravimetric Analysis technique.
Usually, crucibles with a hole are used in DSC analysis to avoid overpressure inside the crucible in the case of evaporation of components. This overpressure could lead to a leakage of the sample inside the DSC oven, which could break the components of the equipment. The manufacturer of DSC equipment recommends using these crucibles with a hole in the lid.
In this case, MMA could evaporate during the DSC scan and, to avoid any accident, we used crucibles with a hole. However, we were not interested in this evaporation, but in the polymerization process and, for this reason, we only analysed DSC curves.
- Which Model-Free method was selected for the analysis? The authors should provide the mathematical expression that describes this method.
In our case, STARe software applies the needed equations to obtain activation energy. This model is based on the work of Vyazovkin et al. (Vyazovkin S, Wight CA. Model-free and model-fitting approaches to kinetic analysis of isothermal and non-isothermal data. Thermochim Acta. 1999;340–341:53–68; https://doi.org/10.1016/S0040-6031(99)00253-1).
The reference has been added to the text.
- Line 321: the monomer addition resulted in a decrease and not increase in polymerization rate, judging from the slope of R’ in figure 5. Α plot of conversion vs time would be more helpful to interpret. In line 350 for instance, it is correctly stated that a change in the slope of the R’+GO curve could indicate a change in polymerization rate towards a faster mechanism.
The addition of monomer slightly reduced the polymerization rate, but the temperature needed to start the polymerization was lower and the initial activation energy was also lower. During the propagation and termination stages, the activation energy is similar to the resin. Therefore, it could be concluded that MMA catalyzed the first stage of the reaction. It has been clarified in the text.
- According to 2020 ICTAC recommendations, Eα can be taken as constant if the difference between the maximum and minimum value is <10–20 % of the average Eα value within α = 0.1–0.9 range. Does this apply for the limits regarding the average value of activation energy that the authors calculated? If so, then activation energy increase or decrease below 10% and above 90% conversion might be inappropriately associated to chemical procedures. This applies for all figures 6,8,10 and 12.
In this work, the reaction that we have analysed have three different stages with different mechanisms. The first stage is the creation of free radicals by heat or UV, which occurs at lower conversion degrees (<10%). Therefore, we need to analyse the curve from 2% of conversion to study the effect of MMA and GBN in this first stage.
For high conversion degrees, the activation energy gives us information about the termination stage, which depends on the crosslinking degree and the availability of free radicals.
For this reason, in Table 12 we considered initialization in 2% of conversion, propagation in 50% and termination in 90%.
- Error bars should be depicted along with activation energy in all figures 6,8,10 and 12.
Curves were obtained from a high number of points and error bars would make it difficult to extract conclusions from the curves. Each test was done three times and the conversion curves were similar in all cases, with an error lower than 1%. In the text the following sentences have been added:
“Each scan was carried out at least three times.” (Experimental procedure)
“Each curve was obtained three times and the error was estimated to be lower than 1%.” (Activation energy figures).
- How do the authors come up with the conclusion about which chemical reaction takes place at different conversion values by solely applying model free kinetics? E.g. in line 335 it is stated that activation energy increases from around 85% of conversion because of diffusion control. Such an assumption cannot rely on one single reference, rather proved through the kinetic analysis. If for instance the authors could track the glass transition temperature throughout all the polymerization range, or the conversion at gelation they could attribute diffusion to vitrification or gelation events.
The polymerization process is known and, from activation energy curves, it is possible to distinguish between the different stages that take place. Termination stage usually requires higher activation energy because the reaction is controlled by the diffusion process because the access to free radicals is more difficult due to the high crosslinking of the polymer chains at this high conversion degrees. It can be seen in the work carried out by Paz et al (Graphene Oxide and Graphene Reinforced PMMA Bone Cements: Evaluation of Thermal Properties and Biocompatibility, Materials 2019, 12, 3146). This reference was added in the manuscript (Section 2.7).
Vitrification and gelation would be seen as an endothermic peak near the glass transition temperature followed by the exothermic polymerization peak, and it was not found in our curves.
- What heating rate did the authors use to evaluate the enthalpy values in tables 6, 7, 9 and 10? Moreover, enthalpy values that correspond to each different heating rate should be included in all the tables mentioned above.
Enthalpy values are not dependent on the heating rate. Values shown in tables 6, 7, 9 and 10 are the mean values of the three heating rates used.
- Table 7 and line 343: how do the authors explain the lower enthalpy values and the possibly lower polymerization degree for the system R’+G, since polymerization peak temperature is higher at all three heating rates, compared to the other two systems? Shouldn’t higher temperature lead to higher crosslinking degree?
Higher peak temperatures could be associated with a retardation effect in the polymerization reaction. Together with this retardation effect, R’+G mixture showed lower enthalpy, which could be due to a lower polymerization degree. Besides, activation energy was higher than R’ and R’+GO. All these findings suggested that R’+G mixture achieved lower polymerization degree than the other studied mixtures, as explained in the manuscript.
- Why did the authors choose to study the residual polymerization of UV treated samples with DSC, rather with FTIR since the last method was the one selected to calculate the extent of curing after UV treatment (table 8)? Moreover, how do they explain the similar trend in conversion curves (figure 9) between R and R’, considering the second has already reached approximately 40% of conversion at the initial stage of DSC measurement?
The objective of using DSC instead of FTIR to study the polymerization process was to obtain and analyze the activation energy during the whole process to explain the changes in polymerization rate and mechanism.
Regarding the trend in conversion curves of R and R’, it seemed that the polymerization mechanism was different for thermally-triggered polymerization due to the presence of different kind of acrylate monomers. However, when the polymerization was triggered by UV, the polymerization mechanism was the same, being less important the nature of the monomers. Free radicals were already formed by UV exposition.
- What is the explanation of similar glass transition Tg values between all the examined systems (table 11) after evaluating such differences in enthalpy values and peak temperatures? Moreover, why do the authors compare crosslinking and polymerization degree of the different structures (lines 343, 386), when similar Tg values are detected between them?
The polymerization process was different, and the crosslinking degrees were different in each stage for each kind of nanocomposite. However, when the resin was completely polymerized, the crosslinking degree was similar in all cases, as can be seen in the Tg values.
- Finally, the text, although well written, should be once again checked for spelling and expression at some points. E.g. in line 49 of the abstract the expression ‘being necessary an adaptation of the printing parameters to allows the printability of these nanocomposites’ should be replaced by ‘while an adaptation of the printing parameters is necessary in order to allow the printability of these nanocomposites.
The text has been checked.
Reviewer 2 Report
This mansucript reported kinetic load with 2d materials for AM. The logic is reasonable. I recomment to accept after the following isssue are solved.
1, What is the reason for 405 nm in Table 4? any reason to choose 405 nm?
2, Any reason for GO and G in Table 5? What is the mechanism?
3, What is the novelty of this manuscript?
4, What is the reason to make UV in this manuscript in Figure 11?
5, The author can cite more papers to solid the background, such as Micromachines. 11, 7, 541, 2020.
6, What is the purpose of this manuscript? The authors can make it more clear
Author Response
Dear Editor and Reviewer:
We are sending you the revised version of the manuscript entitled:
"Polymerization Kinetics of Acrylic Photopolymer loaded with Graphene-Based Nanomaterials for Additive Manufacturing” (Manuscript ID: nanomaterials-2034991), prepared by S. Lopez de Armentia, J. Abenojar, Y. Ballesteros, J.C. del Real, N. Dunne, E. Paz.
We thank the reviewers for their careful reading of the manuscript and their valuable comments that have helped us to improve it.
We hope to have answered all the queries raised by the reviewer and the editor and that the manuscript is now acceptable for publication in Nanomaterials. The reviewer’s comments followed by the corresponding answers are shown as follows:
This mansucript reported kinetic load with 2d materials for AM. The logic is reasonable. I recomment to accept after the following isssue are solved.
1, What is the reason for 405 nm in Table 4? any reason to choose 405 nm?
SLA printers usually use two different wavelengths: 365 or 405 nm. The laser of Formlabs printer used in this study produces light of 405 nm of wavelength.
This sentence was added in the text: “The wavelength was selected as a function of the wavelength of the printer’s laser.”
2, Any reason for GO and G in Table 5? What is the mechanism?
The reduction in Dp was probably due to the high absorbance of the resin loaded with the GBN. If GBN absorbs part of the UV light, it will reach lower penetration depth. In the case of Ec, as seen later in polymerization kinetics, GO catalyses the polymerization reaction, whilst G inhibits it. The following sentence has been added to the manuscript: “This effect could be due to the high absorbance of R’+GBN because of the change in the colour of the resin”.
3, What is the novelty of this manuscript?
The following sentence has been added to the text: “To the best of our knowledge, in the literature, only hypotheses have been made regarding the effect of nanofillers on the properties and the polymerization of photocurable resins."
4, What is the reason to make UV in this manuscript in Figure 11?
This resin was especially formulated to obtain structures by stereolithography, where the polymerization will be through UV exposition. Therefore, it is interesting to understand how MMA and GBN affect to the polymerization via UV. This application was especially important in the biomedical field, like odontology.
5, The author can cite more papers to solid the background, such as Micromachines. 11, 7, 541, 2020.
We cannot find the paper that the reviewer recommended. Please, check the number of the paper. The volume 11, issue 7 has papers from 626 to 707. Paper number 541 is in the issue 6 and it is not related with this work.
6, What is the purpose of this manuscript? The authors can make it more clear
The following sentence was added to the introduction: “Therefore, the purpose of this research is to establish a foundation to understand the effect of GBN on the polymerization of photocurable resins. Besides, a solution to obtain homogeneous dispersion of GBN that does not affect negatively the polymerization process was proposed.”
Reviewer 3 Report
This work offered by S. Lopez de Armentia and co-workers is some interesting. In terms of the contents, this work could be published after fairly extensive modifications as followed:
(1). Figures 1 and 2 should be deleted.
(2) The whole article needs careful typesetting and is in a mess.
(3). Lack of basic performance characterization, such as thermogravimetry, elemental analysis, infrared…..
(4). There are some typos and grammatical errors, the authors must make an overall revision to improve the readability. For example,
(a) “The objective was to investigate the influence that the incorporation of GBN and MMA have…” should be “The objective was to investigate the influence that the incorporation of GBN and MMA has…”
(b), “…some studies have reported how 3D printed components fabricated from a GBN-based composite material have”, why two “have” exist?
(c) as for “…analyzing tensile strength it was observed…”, very unprofessional expression, more like pulling the strings
Author Response
Dear Editor and Reviewer:
We are sending you the revised version of the manuscript entitled:
"Polymerization Kinetics of Acrylic Photopolymer loaded with Graphene-Based Nanomaterials for Additive Manufacturing” (Manuscript ID: nanomaterials-2034991), prepared by S. Lopez de Armentia, J. Abenojar, Y. Ballesteros, J.C. del Real, N. Dunne, E. Paz.
We thank the reviewers for their careful reading of the manuscript and their valuable comments that have helped us to improve it.
We hope to have answered all the queries raised by the reviewer and the editor and that the manuscript is now acceptable for publication in Nanomaterials. The reviewer’s comments followed by the corresponding answers are shown as follows:
This work offered by S. Lopez de Armentia and co-workers is some interesting. In terms of the contents, this work could be published after fairly extensive modifications as followed:
(1). Figures 1 and 2 should be deleted.
They have been deleted.
(2) The whole article needs careful typesetting and is in a mess.
To make it easier to read and follow the paper structure, a scheme has been added to the manuscript.
(3). Lack of basic performance characterization, such as thermogravimetry, elemental analysis, infrared…..
FTIR spectrum of the acrylic resin has been added to characterize the polymer. We don´t have thermogravimetry equipment.
(4). There are some typos and grammatical errors, the authors must make an overall revision to improve the readability. For example,
(a) “The objective was to investigate the influence that the incorporation of GBN and MMA have…” should be “The objective was to investigate the influence that the incorporation of GBN and MMA has…”
(b), “…some studies have reported how 3D printed components fabricated from a GBN-based composite material have”, why two “have” exist?
(c) as for “…analyzing tensile strength it was observed…”, very unprofessional expression, more like pulling the strings
The text has been reviewed to improve readability.
Round 2
Reviewer 1 Report
The current work entitled Polymerization Kinetics of Acrylic Photopolymer loaded with Graphene Based Nanomaterials for Additive Manufacturing, is a research with considerable experimental work associated to the polymerization kinetics of an acrylic photocurable resin, before and after reinforcement with graphene based nanomaterials (GBN) and under the effect of MMA addition. The study could be published after considering the following:
1. According to comment 2 from previous review stating that: DSC curves (heat flow vs temperature) should be included for the tested samples (the first and second run) under all estimated different heating rates (5,10 and 20oC/min), the authors have provided supplementary data where the first heating scans for all tested systems and heating rates are depicted. However, the results are quite problematic for the following reasons: a) the starting and the final time for the subtraction of the baseline are inside the polymerization b) it seems that the shape of the baseline is not the same for all the experiments. For instance in fig. S1, b (green) the baseline is totally wrong and the calculated enthalpy maybe it is at least 20% lower than the real one. For the above reasons all the calculated enthalpies present significant errors. The authors must extent the area before and after the polymerization peak in order to have a real linear part and then they must decide which type of baseline line it is better to use, often the tangential type is the most appropriate. The type of the baseline must be used for all the studied samples.
2. Moreover, in figures S2 c and d from the supplementary data, there seems to be an endothermal glass transition for the heating rate of 20 oC/min right before the polymerization curve, which is not evidenced for the other two heating rates, while it should have, why is that?
3. Referring to the previous comment, the second reheating scans could be depicted as well, in order to firstly confirm successful polymerization and secondly to display the glass transition of the polymerized samples.
4. Finally, in some cases, e.g. figure S1 a and b and figure S2 c, all referring to a heating rate of 5oC/min, there seems to appear a double peak in the curing curve, in contrast to the higher heating rates of the same system. Authors should explain why this is happening and whether the polymerization mechanism varies with the heating rate.
Author Response
Dear Editor and Reviewer:
We are sending you the revised version of the manuscript entitled:
"Polymerization Kinetics of Acrylic Photopolymer loaded with Graphene-Based Nanomaterials for Additive Manufacturing” (Manuscript ID: nanomaterials-2034991), prepared by S. Lopez de Armentia, J. Abenojar, Y. Ballesteros, J.C. del Real, N. Dunne, E. Paz.
We thank the reviewers for their careful reading of the manuscript and their valuable comments that have helped us to improve it.
We hope to have answered all the queries raised by the reviewer and the editor and that the manuscript is now acceptable for publication in Nanomaterials. The reviewer’s comments followed by the corresponding answers are shown as follows:
The current work entitled Polymerization Kinetics of Acrylic Photopolymer loaded with Graphene Based Nanomaterials for Additive Manufacturing, is a research with considerable experimental work associated to the polymerization kinetics of an acrylic photocurable resin, before and after reinforcement with graphene based nanomaterials (GBN) and under the effect of MMA addition. The study could be published after considering the following:
- According to comment 2 from previous review stating that: DSC curves (heat flow vs temperature) should be included for the tested samples (the first and second run) under all estimated different heating rates (5,10 and 20oC/min), the authors have provided supplementary data where the first heating scans for all tested systems and heating rates are depicted. However, the results are quite problematic for the following reasons: a) the starting and the final time for the subtraction of the baseline are inside the polymerization b) it seems that the shape of the baseline is not the same for all the experiments. For instance in fig. S1, b (green) the baseline is totally wrong and the calculated enthalpy maybe it is at least 20% lower than the real one. For the above reasons all the calculated enthalpies present significant errors. The authors must extent the area before and after the polymerization peak in order to have a real linear part and then they must decide which type of baseline line it is better to use, often the tangential type is the most appropriate. The type of the baseline must be used for all the studied samples.
In all cases, the baseline was done with the same type of baseline: spline. As three scans were done for each sample, the curve shown was changed.
- Moreover, in figures S2 c and d from the supplementary data, there seems to be an endothermal glass transition for the heating rate of 20 oC/min right before the polymerization curve, which is not evidenced for the other two heating rates, while it should have, why is that?
The small peak that appeared before polymerization peak was due to the crosslinking of the UV-polymerized resin. It was previously studied by the authors (S. Lopez de Armentia et al. 3D printing of a graphene-modified photopolymer using stereolithography for biomedical applications: a study of the polymerization reaction. International Journal of Bioprinting 8 (1), 2022).
It only appeared for the fastest scan because in DSC when the rate is reduced the peaks appeared less marked and this is a small peak.
- Referring to the previous comment, the second reheating scans could be depicted as well, in order to firstly confirm successful polymerization and secondly to display the glass transition of the polymerized samples.
Second scans were added to Supplementary files. However, in some cases and especially for UV-triggered polymerized samples, it was difficult to find the Tg because of the low delta Cp, which could be due to the high crosslinking degree of UV-triggered polymerized samples. However, the software was able to measure the Tg, despite the small step found.
This sentence was added to the text: “The thermograms used to measure glass transition temperature are shown in Figure S3. They show the successful polymerization in all cases.”
- Finally, in some cases, e.g. figure S1 a and b and figure S2 c, all referring to a heating rate of 5oC/min, there seems to appear a double peak in the curing curve, in contrast to the higher heating rates of the same system. Authors should explain why this is happening and whether the polymerization mechanism varies with the heating rate.
The polymerization mechanism followed by polymers that cures via free radicals shows different rates at the different stages (i.e. initialization, propagation and termination). This difference in polymerization rate can be seen in the thermogram as an asymmetric polymerization peak, which could be separated in two different peaks by deconvolution. Depending on the heating rate, sometimes it is possible to see the two different peaks, but it could be associated with the accuracy of the measurement. To apply MFK model, we need to have only one polymerization peak. For this reason, the deconvolution was not done.
The presence of two peaks could be associated with the autoacceleration of the reaction in the propagation stage, as studied by Jaso et al (V. Jaso et al. Analysis of DSC curve of dodecyl methacrylate polymerization by two-peak deconvolution method. Journal of Thermal Analysis and Calorimetry 101, 2010).
It was added to the manuscript: “The asymmetry of the polymerization peaks and the possible appearance of two different peaks in some nanocomposites and rates could be due to the different rates at the different stages due to the autoacceleration in the propagation stage.”
Reviewer 2 Report
The author did answer all my issues. I recommend to publish in the current form
Author Response
Many thanks for your time and your contribution to improve the manuscript.
Reviewer 3 Report
The revision is suitable for direct acceptance.
Author Response

(The authors gave the same response as above.)

Round 3
Reviewer 1 Report
The authors study the Polymerization Kinetics of Acrylic Photopolymer loaded with Graphene Based Nanomaterials for Additive Manufacturing. The authors reply and the new version of the manuscript does not satisfy the reviewer comment.
The basic argument refers to the calculations of the polymerization enthalpy. In the previous report I mention that “the enthalpy results are quite problematic for the following reasons: a) the starting and the final time for the subtraction of the baseline are inside the polymerization b) it seems that the shape of the baseline is not the same for all the experiments. For instance in fig. S1, b (green) the baseline is totally wrong and the calculated enthalpy maybe it is at least 20% lower than the real one. For the above reasons all the calculated enthalpies present serious errors. The authors must extent the area before and after the polymerization peak in order to have a real linear part and then they must decide which type of baseline line it is better to use, often the tangential type is the most appropriate”.
The authors do not reply for this. Also, they did not give the enthalpy values for every heating rate just saying that they are the same. The enthalpy values are same if we subtract baseline is such a way in order to take the values which the authors want.
Since the calculated values of the enthalpy are totally wrong all the following calculations are also totally wrong.
There is no need to present my arguments to many other authors reply, most of them are wrong or there is no reply in reality.